# Thermodynamic Insights into the Oxidation Mechanisms of CrMnFeCoNi High-Entropy Alloy Using In Situ X-ray Diffraction

**DOI:** 10.3390/ma16145042

**Published:** 2023-07-17

**Authors:** Muhammad Arshad, Saira Bano, Mohamed Amer, Vit Janik, Qamar Hayat, Yuze Huang, Dikai Guan, Mingwen Bai

**Affiliations:** 1Centre for Manufacturing and Materials, Coventry University, Coventry CV1 5FB, UK; arshadm15@uni.coventry.ac.uk (M.A.); amerm5@uni.coventry.ac.uk (M.A.); ac6600@coventry.ac.uk (V.J.); hayatq@uni.coventry.ac.uk (Q.H.); ad9358@coventry.ac.uk (Y.H.); 2Department of Chemical Engineering, University of Engineering and Technology, Peshawar 25000, Pakistan; saira.bano@uetpeshawar.edu.pk; 3Department of Mechanical Engineering, University of Southampton, Southampton SO17 1BJ, UK; dikai.guan@soton.ac.uk

**Keywords:** high-entropy alloy, oxidation, in situ X-ray diffraction, ThermoCalc, CALPHAD method

## Abstract

This paper utilizes in situ X-ray diffraction (XRD) to investigate the high-temperature oxidation behaviour of CrMnFeCoNi high-entropy alloy (HEA). We found that (1) Mn is the major oxide-forming element in both vacuum and air environments, leading to the formation of non-protective oxides that deplete the bulk alloy of Mn; (2) no oxides like Cr_2_O_3_, Fe_2_O_3_, or Fe_3_O_4_ were observed during the high-temperature oxidation behaviour of CrMnFeCoNi, which contradicts some previous studies on the isothermal oxidation of CrMnFeCoNi HEA. We also analysed and compared the experimental results with thermodynamic calculations by using ThermoCalc version 2022b software following the CALPHAD method. ThermoCalc predicted spinel oxide in a vacuum environment, along with halite oxides observed in experimental results; also, in an atmospheric environment, it predicted only spinel, indicating the need for further investigation into factors to validate the thermodynamic predictions. Our study shows that the in situ HTXRD technique is a powerful tool to accurately identify time–temperature-dependent phase formation/transformation for studying oxidation behaviours and understanding oxidation mechanisms in HEAs.

## 1. Introduction

First reported in 2004 [1], high-entropy alloys (HEAs) are alloys comprising five or more elements in equal or nearly equal molar fractions, with each element present at a mole fraction of approximately 5–35 at.% [2]. The presence of multiple elements leads to a high configurational entropy, resulting in solid-solution structures with face-centred cubic (FCC) and body-centred cubic (BCC) lattices, rather than intermetallic compounds [3]. In recent years, HEAs have garnered significant consideration in the field of materials science because of their exceptional performance characteristics, including high strength [4], corrosion resistance [5], soft magnetic properties [6], high hardness [7], wear resistance [8], fatigue resistance [9], and softening resistance at high temperatures [10]. The superior high-temperature behaviour of HEAs can be attributed to four underlying effects: the high entropy effect, the sluggish kinetic diffusion effect, the severe lattice distortion effect, and the cocktail effect [11].

Chen et al. [12] assessed the tensile performance of the equiatomic NbMoCrTiAl HEA at temperatures ranging from 800 °C to 1200 °C. Results signified an increase in ductility with rising temperature, with the specimen retaining high-yield strength up to 800 °C. Moreover, the high-temperature oxidation behaviour of HEAs is crucial in high-temperature environments, as severe surface oxidation can have a drastic impact on their mechanical properties [13]. Jin et al. [14] studied the high-temperature wear properties of FeNiCoAlCu HEA coatings, discovering effective thermal stability at 780 °C and outstanding wear properties because of the formation of oxide films on the surface. Kai et al. [15] investigated the oxidation properties of FeCoNiCrSix (x = 0.25, 0.5, and 1.0) HEAs at temperatures ranging from 700–900 °C. Findings indicated that the oxidation rate increased with temperature, and the Si content influenced the oxidation rate, with a minimum Si content leading to a higher oxidation rate. Butler et al. [16] explored the effect of Al content on the oxidation behaviour of AlxCoCrFeNi HEA and found that Al_2_O_3_ was the main oxidation product with some Cr_2_O_3_. Al_2_O_3_ film density and uniformity improved with increasing Al content, resulting in higher high-temperature oxidation resistance. Liu et al. [17] studied the oxidation behaviour of four different HEAs, including Al_0.5_NbCrMoTi, Al_0.5_NbCrMoV, Al_0.5_NbCrMoTiV, and Al_0.5_Si_0.3_NbCrMoTiV, finding that Ti and Si addition improved the oxidation resistance of HEAs, while oxidation behaviour degraded with V addition.

An important high-entropy alloy, CrMnFeCoNi is significant due to its unique composition and exceptional properties. This alloy exhibits remarkable mechanical, thermal, and functional characteristics, making it a promising candidate for a wide range of applications [1,18]. By understanding the fundamental principles underlying the behaviour of CrMnFeCoNi HEA, including its high-temperature oxidation resistance, and mechanical/thermal stability, researchers can unlock its full potential for applications in aerospace, energy generation, and nuclear power. Moreover, CrMnFeCoNi HEA serves as a model system for exploring the broader field of high-entropy alloys, paving the way for the development of novel materials with tailored properties for a wide range of applications, from aerospace to energy and beyond. In this regard, understanding the oxidation behaviour of CrMnFeCoNi HEA is one of the important properties on the checklist for any candidate alloy to be approved for high-temperature application. A large amount of literature has been published on the oxidation of CrMnFeCoNi HEA. These studies have analysed the oxidation product at the end of the isothermal oxidation tests conducted in varying atmospheres or pre-treatments of the alloy before oxidation tests. Ye et al. [19] analysed the effect of annealing on the high-temperature oxidation properties of micro-plasma arc-processed CoCrFeMnNi HEA cladding layers at 900 °C. The results indicated that annealing treatment reduced the high-temperature oxidation resistance of the cladding layer. Laplanche et al. [20] investigated the oxidation behaviour of cast CrMnFeCoNi HEA at temperatures ranging from 500 °C to 900 °C and observed that the oxidation products varied at different temperatures, with Mn_2_O_3_ oxide changing to Mn_3_O_4_ between 800 °C and 900 °C. The oxidation rate was linear at the beginning but became parabolic with prolonged oxidation time [19]. Several other studies have been conducted on this topic, but their conclusions remain ambiguous due to variations in findings based on post-oxidation analysis assumptions [21,22,23,24]. Stephan-Scherb et al. [25] compared the oxidation of CrMnFeCoNi HEA and CoCrNi medium-entropy alloy (MEA) in an atmosphere containing water vapour and analysed the oxide composition of the two alloys, illustrating the specific oxide generation processes. In their study, researchers investigated the oxide composition of both alloys and elucidated the distinctive mechanisms underlying oxide formation for each alloy. Doležal et al. [26] approached the problem from a theoretical standpoint, employing density functional theory (DFT) and Monte Carlo calculations (MC) to examine the oxygen-absorption process and the distribution of transferred charge on the surface of high-entropy alloys. Their research yielded valuable insights that can guide the thermodynamic analysis of HEAs during the initial stages of oxidation. Additionally, DFT was used to study the oxidation thermodynamics of Nb–Ti binary alloys and to calculate the oxide composition and structural changes during the oxidation process, offering an effective method for analysing oxidation processes in HEAs [26]. To summarize, the mechanisms of high-temperature oxidation in CrMnFeCoNi system reported in references [13,27,28,29] were mainly based upon surface analysis of the reaction products retained after tests, which is insufficient because great emphasis has been placed on post-mortem analyses of reaction products analysed at ambient temperature, or else assumptions made via the analysis of the temperature dependence of phases at different temperatures. Clearly, this introduces a degree of uncertainty, since one cannot be completely sure of the underlying physical mechanisms which are imperative on the critical scales: that of the activity of elements, transformation temperatures, and vapour pressure at the same time. Time–temperature-resolved in situ XRD observations would be invaluable in circumventing this ambiguity and in elucidating the chemical processes giving rise to the oxidation of CrMnFeCoNi alloy.

In this study, CrMnFeCoNi HEA behaviour responding to changes in temperature and environmental factors such as the partial pressure of oxygen has been investigated by in situ XRD at high temperatures up to 1100 °C. In situ thermal observations of materials with the high-temperature XRD provide unique information such as temperature-dependent phase transformations, types of new phase formation, and the start and finish temperatures of competing phases in an alloy system [30]. In situ HTXRD is instrumental in revealing the real-time of oxide formation as well as the type and temperature at which various kinds of oxide form when metals are exposed to high temperatures in air or in another simulated environment. Our HTXRD analysis provided insight into the thermal stability, phase transitions, and high-temperature oxidation behaviour of the CrMnFeCoNi alloy samples, which provides significant guidance in the development of high-performance alloys for demanding applications. The surface and cross-sectional microstructure, elemental distributions, and oxide compositions have been examined to further understand the evolution of the oxides. Experimental results findings have been compared with thermodynamic calculations performed using ThermoCalc following the CALPHAD method. This produced a deep appreciation of the factors causing oxidation in CrMnFeCoNi HEA at varying temperatures and environments.

## 2. Materials and Methods

### 2.1. Materials Preparation

CrCoFeMnNi equiatomic alloy (i.e., 20 at.% each) was prepared in the form of rectangular ingot through arc melting and drop casting. The five elements Cr, Co, Fe, Mn, and Ni of 3 N purity were weighed up, and the mixture was loaded into the arc melter. The chamber was vacuumed to ~7 × 10^−4^ Pa and then refilled with inert argon gas to a pressure of ~8.4 × 10^4^ Pa before melting the batch in a water-cooled copper hearth. The cast alloy was re-melted multiple times and, for thorough mixing of the elements, highly effective electromagnetic stirring was used to promote homogeneity before dropping into the rectangular Cu mould. Melted alloy drops were cast into a mould with a rectangular cavity. The drop cast ingot was annealed at 900 °C for 48 h and cold rolled to a thickness of 1.6 mm. The overall composition of the alloy (nominal and experimental) as shown in Table 1 was determined by energy dispersive X-rays.

### 2.2. Materials Characterisation

The rectangular alloy specimen was machined for samples of dimensions (10 mm × 10 mm × 1.6 mm) using a Struers Secotom-50 machine equipped with a SiC cutting wheel at slow speed (0.05 mm/s) to avoid the introduction of strain deep into samples. Samples were prepared for various microstructural characterisation analyses. Cross-sections of the cut samples were metallographically prepared through a combination of fine grinding which was followed by polishing, first using alumina suspension (particle size 300 nm) and then colloidal silica (particle size 50 nm) on Buehler AutoMet^TM^ 300 (Lake Bluff, IL, USA). Highly polished samples were carbon sputtered (Quorum-Q150T ES, Lewes, UK) to reduce sample charging during SEM analysis.

#### 2.2.1. Microstructure Analysis

The microstructural characterisation of the samples was conducted using the Carl ZEISS Gemini SEM (Zeiss Sigma 500 VP, Oberkochen, Germany) which is equipped with a field emission gun and an EDX detector. This high-resolution imaging system allowed for the detailed examination of the samples’ surface and cross-sectional features. Images of the samples’ surface and cross-sections were taken with SEM to gain insight into microstructure. For chemical composition of the samples line, and area maps were obtained using EDX analysis at an accelerating voltage of 20 kV. The imaging and chemical analysis were performed in parallel to identify any chemical inhomogeneities present in the samples.

#### 2.2.2. In Situ High Temperature X-ray Diffraction (HTXRD)

Samples of desired smooth and flat surfaces for in situ HTXRD analysis were sectioned from the prepared alloy with the dimensions 10 mm × 10 mm × 1.6 mm and followed a comprehensive polishing process. The process involved a mechanical polishing step utilizing silicon carbide paper, followed by a mirror polishing step with aid of a 1 μm diamond suspension. The HTXRD analysis was performed using a state-of-the-art Bruker Advance D8 diffractometer, which was equipped with an Anton Paar Domed Hot Stage for Four-Circle Goniometers DHS 1100, as shown in Figure 1. Two different experimental set-ups were designed for both non-ambient experiments. The sample was placed on an AlN baseplate within the heating stage/furnace to avoid contact during oxidation, and a vacuum of approximately ~4.2 × 10^−6^ atm was established using a vacuum pump for HTXRD oxidation analysis of the sample in a vacuum.

In situ XRD tests were conducted from 25 °C to 1100 °C with an interval of 50 °C after a holding time of 5 min at every set point for stabilisation on a Bruker D8 Advance diffractometer using CuKα radiation (λ = 0.154 nm) with a Ni-filter, and an applied voltage of 40 kV and current of 40 mA. The step size of 0.02° was employed, and the scan was taken in 2θ range of 30–80°. The graphical representation of the experiment set-up is depicted in Figure 2, and the temperature profile and XRD scans were carried out using the established protocols.

### 2.3. Thermodynamic Calculations for High-Temperature Oxide Formation

The CALPHAD technique is used to minimize the global Gibbs free energy of a system as a function of temperature and composition and hence applied to CrCoFeMnNi HEA to predict the phase equilibria. A one-dimensional step diagram was selected to visualize the phase equilibria of the CrCoFeMnNi alloy’s phase fraction with temperature. This equilibrium diagram represents the phases that form in a situation when the alloy is held at an annealing temperature for a sufficiently long time to allow complete diffusion of atoms to rearrange themselves to form the most thermodynamically stable configuration with optimum Gibbs free energy.

Thermodynamic calculations were performed for oxide scale formation on the CrMnFeCoNi alloy at a high temperature as a function of oxygen partial pressures at different temperatures. A wide explanation of the modelling background for Thermo-Calc^®^ can be found in the reference [31]. A commercially available TCS Ni-based Superalloys Database (TCNi12) was used for the equilibrium computations [32] instead of HEA6v due to the unavailability of oxygen in the HEA database. The utilisation of Thermo-Calc calculations proves to be a beneficial tool for investigating the selective oxidation characteristics of high-temperature alloys, particularly in the case of complex multicomponent alloys [33,34]. All calculations for oxidation were performed using equilibrium conditions, such as experimental, to match the in situ HTXRD oxidation test used in the present study. The effective oxygen partial pressure (*p*O_2_) is considered as a variable and represented by chemical activity (a, the unit is 1) ranging from 1 × 10^−14^ to 1.0 atm. Table 2 presents the input of conditional parameters required for calculating the phase composition of the oxides formed on the alloys during equilibrium. These parameters play a crucial role in the determination of the resultant oxide phases. However, thermodynamic calculations focus on equilibrium conditions and do not account for the kinetics of reactions. In high-temperature oxidation, the rate of oxidation and the formation of oxide layers are influenced by factors such as diffusion, nucleation, and surface reactions. These kinetic effects are not considered in thermodynamic calculations and can lead to discrepancies between predicted and experimental results. Furthermore, the accuracy of all these properties depends highly on the quality of the databases that, in turn, ultimately rely on the available experimental data for optimizing the model parameters and validation. The quality and reliability of the data can vary, especially for less studied systems or at high temperatures. So, in certain situations, inaccurate or incomplete thermodynamic data can lead to deviations between calculated and experimental results [35,36].

## 3. Results

### 3.1. Microstructural Analysis of CrMnFeCoNi High-Entropy Alloy

A representative low-magnification SEM image of the materials in its as-homogenized state is shown in Figure 3. CrCoFeMnNi HEA exhibits coarse equiaxed grain structure usually with a grain size up to 0.1 mm after homogenisation. Many of the grains are observed to have annealing twins in a recrystallized condition. In both homogenized and non-homogenized states examined, the presence of visible pores were found; these pores appear to represent casting porosity or holes left behind from particles that were removed during the polishing stage. Also, noticeable whole particles were observed, typically found in grains. By using EDX to study a few of these particles, it was discovered that they were abundant in Cr, Mn, and O, signalling that they are oxides, possibly of type MnCr_2_O_4_ (based on their EDX spectra). Given that this type of alloy has previously been reported of such inclusions, their presence is not surprising [37,38,39].

Figure 3c shows SEM-EDS map elemental analysis of the initial, as prepared, CrCoFeMnNi HEA sample. The EDS map (Figure 3c presents no micro-level defects such as sigma phase, and all elements were distributed equally with no segregation or clustering, which supports the specimen status as single-phase solid solution. Similarly, only an FCC solid-solution phase was found through X-ray diffraction with peaks at 43.57°, 50.69°, and 74.56° from (111), (002), and (022), respectively, with no ordered structure, which is commonly found in CrCoFeMnNi alloy. These EDX results are consistent with work conducted [39] at a similar length scale, which reports that CrMnFeCoNi HEA is a single-phase solid solution after homogenisation at an elevated temperature and even after plastic deformation.

### 3.2. HTXRD in Vacuum

To examine the temperature dependence of the oxidation mechanism of CrMnFeCoNi HEA, in situ X-ray diffractograms were obtained in a vacuum while heating the sample to 1100 °C at a heating rate of 15 °C/min. The selected 2θ range from 30–80° was carefully monitored during measurement, as it contains the reflections of the oxidation species that are more likely generated during the oxidation process. Important diffractograms that contain information about thermal stability and new phase formation in CrCoFeMnNi alloy during the oxidation process are shown in Figure 4a. There was no change in the diffraction pattern from room temperature up to 700 °C; hence, diffractograms obtained from 25 °C to 700 °C are not included in Figure 4a. The phase composition of the specimen surface appears to exhibit change as shown by XRD scan data at a temperature of 700 °C with the appearance of small new peaks. These patterns clearly indicate that the homogenized alloy is not stable at intermediate temperatures and can undergo oxidation. The new peaks during phase identification were found to be MnO, and the substrate material conserved its phase stability throughout up to 1100 °C.

Figure 4b shows the densitometric view of HTXRD (high-temperature X-ray diffraction) recorded during in situ XRD shows the intensity of the X-ray diffraction pattern as a function of temperature. It shows the change in diffraction intensity of a specific crystallographic plane and set of planes, as the temperature is varied in an in-situ experiment. This densitometric view shows information about the structural changes and phase transformations that occurred in the sample when subjected to high temperatures. For a better understanding of changes in the densitometric view, a respective XRD pattern is provided for identifying the initiation and transformation of new phases.

Figure 5a,b shows surface morphologies of the sample used for in situ HTXRD oxidisation during the experiment when heated until 1100 °C in a vacuum. The oxidized surface topography is completely different from the specimen prior to use in HTXRD (Figure 3). A uniform thin oxide scale of greenish-like colour formed as an oxide surface layer with a small granule-like surface structure (Figure 5a,b). The thickness of the oxide layer formed in a vacuum is smaller than that formed in air using HTXRD. An enlarged image of the formed scale in Figure 5b shows numerous sugar-like granules distributed evenly on the oxidized surface. EDS analysis of the granules shows that the main component elements of the nodules area are Mn and O. Also, the EDS line scan shows peaks in the intensities of Mn along the oxide-substrate length, shown in Figure 5d, where a high count of Mn in the oxide scale was observed and then dipped near the surface. Furthermore, the EDS mapping results show that the oxide scale formed in a vacuum is solely an Mn-oxide layer with no other type of oxides.

### 3.3. HTXRD in Air

An in situ XRD study on CrMnFeCoNi alloy in air is shown in Figure 6. The phase composition on the surface of the specimen changed during in situ HTXRD, which resulted in variation of XRD patterns obtained progressively during HTXRD. Diffractograms obtained from room temperature up to 550 °C show no characteristic change in XRD patterns except for the peak’s movement towards small angles of 2θ, suggesting a change in the substrate FCC phase lattice parameters. At 550 °C, a small new peak appears at 2θ value of 32.71° which, later at a temperature of 650 °C with the appearance of new peaks, was confirmed to be the initialisation of Mn becoming oxidation at 550 °C in air. According to the relative intensity of diffraction peaks, the Mn-rich oxide dominated throughout the in-situ oxidation test. The oxide formed in air was Mn_2_O_3,_ which is different from MnO formed in the vacuum; thus, we concluded that oxygen abundance promotes the formation of Mn_2_O_3_ instead of MnO.

With the increasing temperature, Mn_2_O_3_ continuously forms on the surface through the continuous diffusion of Mn to the surface, and then, at 800 °C, new peaks appear (Figure 6a), which were found to be MnCr_2_O_4_. The intensity of Mn_2_O_3_ peaks weakens at 900 °C compared to its peaks at 850 °C or lower temperatures and finally vanishes after 950 °C. At 950 °C, Mn_3_O_4_ starts to form as small new peaks and as it consumes the peaks belonging to Mn_2_O_3_. At the beginning of oxidation, the main product was Mn_2_O_3_; however, with longer oxidation and exposure time, Mn_2_O_3_ (at a lower temperature) reacts with oxygen to transform into Mn_3_O_4_ (at a higher temperature). Therefore, the disappearance of Mn_2_O_3_ is caused by the rapid consumption of Mn_2_O_3_ for the generation of the mixed oxides of Mn_3_O_4_, MnCr_2_O_4_, and Mn_3_Fe_3_O_8_ under a multiphase effect [40]. In this time- and temperature-dependent phase transformation, five different types of competing oxide phases formed and vanished at different temperatures and times.

Figure 6c shows the densitometric view of the X-ray diffraction pattern, revealing that no changes occurred in the alloy’s structure at the initial temperature. However, at 500 °C, the appearance of the Mn_2_O_3_ phase was observed, indicating the presence of oxide formation in the alloy. As the temperature increased, the appearance and disappearance of other phases were also detected on the densitometric plot. At 1100 °C, the alloy was held for three readings before being cooled to room temperature, and the final XRD scan was taken. By analysing the densitometric view of the HTXRD pattern, we were able to identify the different phases that appeared at different temperatures and track the kinetics of phase transformations in the CrMnFeCoNi alloy.

Figure 7a,b shows surface topographies of specimens oxidized during HTXRD in air with heating from room temperature to 1100 °C. The colour of the sample was dark brown compared to one in the vacuum with a whitish oxide surface. A uniform thick oxide scale formed in air in comparison to the one formed in the vacuum on the naked surface during the oxidation process. The uniform protective oxide scale can inhibit further oxidation of materials underneath the scale. The presence of a uniform and protective oxide scale serves to impede the continued oxidation of the underlying materials. Figure 7b provides an enlarged image depicting numerous granules dispersed on the oxidized surface, representing the process of oxide nucleation and growth.

The EDS map analysis is shown in Figure 8, which clearly shows the stratified oxide layer formed of a Mn-rich top layer and a Cr-rich inner layer; Fe can also be seen as the third element in smaller concentrations, diffusing out from bulk alloy during oxidation. The Mn-depleted zone is also visible in the EDS map. The EDS analysis results of granules (Figure 7b) in the table show that the main component elements of the nodules area are Mn and O, and the atomic ratio of Mn and O_2_ is about 2:1. For the specimen oxidized in air, the surface oxidation (Figure 7) is more serious than that in the vacuum, and the oxidized surface exhibits an inhomogeneous morphology. In addition, some small debris is observed adhering to the oxide scale, indicting a mild spallation of the oxide layer. The EDS result further indicates that major elements constituting oxide scale in air are Mn, Cr, and O_2,_ with a small amount of Fe.

## 4. Discussion

### 4.1. Phase Stability at High Temperature

HEAs produced on casting when exposed to long-term annealing cause decomposition into the traditional multi-phase microstructure. The alloys decomposition on annealing at a higher temperature creates doubt about its applicability for high-temperature application (over 800 °C). Thermodynamic calculations were performed to understand whether the solid solution established when cast is a thermodynamically stable phase at potential use temperatures. Phase equilibria of CrMnFeCoNi alloy are visualized in Figure 9 as a one-dimensional step diagram of phase fractions and a function of temperature. From Figure 9 the critical features of phase equilibria are readily discernible, such as the thickness of the HEA’s single-phase FCC region (solvus to the solidus) at high temperature from solidus at 1320 °C down to about 750 °C. Also, it shows the lower temperature decomposition products stated in some works. At a lower temperature, phase equilibria show the formation of the sigma phase and BCC phase.

The thermodynamic calculations restrict the CrMnFeCoNi alloy’s thermal stability in the temperature range (750–1320 °C) (see Figure 9), which has been confirmed by experimental work [38] on the long-term stability of CrMnFeCoNi alloy high-temperature annealing. Lower temperature decomposition has been reported in [41] with the formation of new phases. One lower temperature phase is the sigma phase that formed below 585 °C and remained stable in temperatures down to 205 °C. The maximum volume phase fraction of the sigma phase of 0.1854 was at 430 °C, and as reported, the sigma phase is usually distributed along the grain boundaries [42]. Formation of the sigma phase was reported by Otto et al., when CrMnFeCoNi HEA was aged for 500 days at 700 °C, while BBC formed at 500 °C, and only FCC was reported to form at 900 °C when held for the same ageing time [43]. In the microstructure of as-cast alloys, there exist inhomogeneities caused by the partitioning of high melting point/slow diffusing elements from fast diffusing/low melting point elements [43]. These inhomogeneities give rise to variations not only in composition but also in the solidus temperature, which represents the temperature at which incipient melting occurs. In Figure 9, an inset of the equilibrium composition of the sigma phase is observed, which has a high concentration of Cr followed by Fe and Mn, also confirmed by experimental work [41]. Some other studies on the long-term stability of the alloy report results as summarised in Table 3. CALPHAD predictions provide information on the phase equilibria at the heat treatment temperature, as well as the solvus and equilibrium solidus temperatures. A notable quantitative agreement is observed between experimental heat treatment temperatures and the predictions from the CALPHAD database. This agreement correctly captures the structural characteristics of HEA and accurately predicts the occurrence of secondary phases.

### 4.2. Oxidation in Vacuum

The formation of MnO was expected, as Mn is the most reactive element in the alloy mixture to react with oxygen. The fast growth of MnO should be kinetically favourable because of its highly unstable nature. The high-temperature oxidation of Mn to MnO, i.e., 700 °C instead of 550 °C (during HTXRD in air), can be explained by lower oxygen partial pressure. With an increase in oxidation temperature and exposure time, MnO phase peaks’ intensity progressively increased, which indicates Mn’s continuous oxidation. Quantitative analysis of the data revealed a gradual increase in the second phase due to the fast formation of MnO. As Mn has a higher affinity for oxygen, with an increase in temperature, Mn preferentially diffuses from inside to the surface and thus reacts with oxygen. The observed trend in the increase in the phase quantity with time and temperature is in good agreement with parabolic kinetics [28]. As shown in Figure 5c, the scale thickness formed was extremely low due to lower oxygen partial pressure. As We et al. [28] reported, the oxidation rate increases with an increase in oxygen partial pressure during the oxidation behaviour of CrMnFeCoNi alloy in varying levels of oxygen partial pressure. The scale formed remained well adherent with the substrate, although some pores were observed inside the scale. Studies [18,44] report MnO, Mn_3_O_4_, Mn_2_O_3_, and MnO as stable four oxide types in the Mn-O oxide system. This study validates MnO formation due to the lower oxygen partial pressure; while the same material when exposed to high temperature in air, Mn oxidized to Mn_2_O_3_ and Mn_3_O_4_.

Figure 5a,b shows the surface morphology of the sample surface oxidized during HTXRD in a vacuum, and the oxidized surface exhibits the morphology of rugged features with some sugar-like granules. A very thin oxide layer (>0.5 μm) forming in the vacuum is obviously due to a very low level of oxygen, and hence no noticeable change in the intensities of the FCC peaks was observed, as shown in Figure 6. The analysis of the oxide-formed layer revealed the formation of MnO having a cubic crystal structure of lattice parameter 4.510 Å. This is possible due to Mn’s high vapour pressure and consuming the available oxygen and no other types of oxides forming during HTXRD in a vacuum. As shown in Figure 5b, an enlarged SEM image shows numerous cubic-like granules distributed evenly on the oxidized surface, which can be regarded as ‘nucleation and growth’ of oxide. The cross-sectional morphologies of the oxide layer and corresponding elemental distribution along the scale thickness direction of the HEA sample are displayed in Figure 5c,d. Also, the EDS line scan confirms that Mn diffuses out from the near surface, as shown in Figure 5d, with a high counts of Mn in the oxide scale and then dipping in near surface. Combined with the XRD results in Figure 4, the oxidized products are MnO with no other oxide type due to the lower oxygen availability.

### 4.3. Oxidation in Air

The presence of a graded oxide layer, comprising outer Mn-rich and inner Cr-rich layers, has been extensively documented as an oxidation phenomenon [13,20]. This phenomenon is closely linked to the differential diffusion rates of the multi-principal elements within the Cantor alloy. Based on the findings reported by Tsai et al. [1] and Kai et al. [28], the order of elements in terms of the decreasing diffusion rate is as follows: Mn > Cr > Fe > Co > Ni. The initialisation and evolution of different oxide phases that were found with XRD scans agree well with these elements’ diffusion sequence order. Mn, Cr, and Fe preferentially diffused outward and became oxidized on the surface of specimens with time and temperature. Furthermore, as the temperature increases, it becomes apparent that the substrate’s FCC phase diminishes, likely due to the concurrent increase in oxide layer thickness, which consequently leads to a weakening of the peak intensity associated with the FCC phase [13]. No protective oxide-like Cr_2_O_3_ formed; instead, MnCr_2_O_4_ formed at 800 °C, adding to the already identified Mn_2_O_3_ oxide formed at 700 °C. Although Cr_2_O_3_ was not identified as an oxide, a previous work [28] on phase equilibrium relations in manganese oxide-Cr_2_O_3_ at a high temperature reported stability of Mn_2_O_3_ up to 877 °C, which agrees reasonably well with the phase transformation temperature of Mn_2_O_3_ to Mn_3_O_4_ at 900 °C (See Figure 6b) [45]. In alloys for high-temperature applications, at least one protective oxide among BeO, Cr_2_O_3_, Al_2_O_3_, and SiO_2_ are desirable to form, and that is why most of the alloys for high-temperature applications add these elements to form an oxide barrier for better oxidation resistance [46]. Previous studies have indicated that Cr_2_O_3_ is the sole stable phase in the Cr-O system, known for its compact structure and superior oxidation resistance [45]. However, it is noteworthy that Cr2O3 was not detected in the in situ XRD analysis conducted for this study. It can be reasonably inferred that non-existence of a Cr_2_O_3_ layer caused further outward diffusion of Mn, leading to the formation of Mn_3_O_4_, as can be seen from the XRD scan at a temperature of 950 °C.
$$2\mathrm{Mn}+2{\mathrm{Mn}}_{2}O_{3}+O_{2}\to 2{\mathrm{Mn}}_{3}O_{4}$$

However, the list of oxides that were identified is different from most of the studies that analysed the high-temperature isothermal oxidation resistance of CrMnFeCoNi alloy. For instance, other than Mn, no binary oxides such as Cr_2_O_3_ or Fe_2_O_3_ formed as reported in studies [24,27] on the high-temperature oxidation behaviour of Cantor alloy. However, mixed oxide phases were identified during in situ XRD in the form of Mn_3_Fe_3_O_8_, and MnCr_2_O_4_, which signals that the surface and diffused-out Cr and Fe combined with Mn and oxygen to form ternary oxides. The formation of ternary oxides instead of binary oxides could be attributed to the stable binary oxides of Cr and Fe that would form after a long exposure time at the oxidation temperature. This statement suggests that under the specific experimental conditions and timescale of the study, the oxidation process may not have progressed to the point where the stable binary oxides of Cr and Fe could fully form. According to thermodynamic theory [24], the order of Gibbs free energy for oxides formation ΔGf: kJ/mol O_2_) is as follows: Al_2_O_3_ > SiO_2_ > MnO > Cr_2_O_3_ > Mn_3_O_4_ > Mn_2_O_3_ > Fe-oxides > CoO > NiO > Co_3_O_4_, which suggests the affinity of oxygen with metal elements. However, the final oxidation products are determined by the element’s concentration, surrounding atmosphere, and kinetics of the alloy.

The surface morphology of the specimen oxidized during HTXRD in air when heated from room temperature to 1100 °C is shown in Figure 7a,b. The SE-SEM surface image of the rugged surface with granules shows that a small area of the sample surface initially oxidized at 550 °C as can be seen from the XRD pattern, and oxide crystals grew with increasing temperature and exposure time until they coalesced and covered the sample surface [40]. Also, in the enlarged image, the distributed granules on the oxidized surface can be regarded as nucleation and growth of oxide. Mn and Cr were the two main elements, with some amount of Fe that mostly formed the surface oxide layer. From the surface EDS, it was confirmed that major elements in the oxide scale in air are Mn, Cr, and O_2_ with a small amount of Fe. It is worth noting that Mn and Cr depleted in the vicinity of the surface due to the outward diffusion of Mn and Cr by vapor transport. The diffusion rate of Mn is two orders faster than Cr [47], which results in the preferred form of manganese-rich oxides in the outer layer. Also, Mn has the highest vapor pressure among the five alloying elements. It has been reported [24] that the number of pores in the substrate increases with the increasing oxidisation temperature, and the increase in pore quantity may accelerate the vapor transport in the substrate through pores, which promotes the growth of the Mn-rich oxide outer layer [46]. In addition, the thickness of oxide containing Cr increases with increasing temperature. As no Cr-rich oxide of the form Cr_2_O_3_ formed during the oxidation test to work as a barrier and inhibit or slow down the diffusion of Mn element, no protection of the substrate beneath the scale to further oxidation was thus available [48].

Figure 7c shows the cross-section topography after high-temperature XRD oxidation in air. An irregular oxide scale formed and remained adherent to the substrate. The oxide scale thickness is about 1.58, 2.53, and 3.45 μm at different locations, which indicates that oxidation does not occur uniformly on the surface. Furthermore, no evidence of internal oxidation attack along the grain boundaries within the substrate was observed. In addition, no internal oxidation attack in the substrate is seen as having been executed along the grain boundaries. Figure 8 presents a cross-sectional EDS map of the specimen oxidized when heated during in situ XRD. For the specimen oxidized, it was observed that a mixed oxide layer consisting of Mn, Cr, and Fe oxides on the substrate and the oxides were confirmed to be Mn_2_O_3_, Mn_3_O_4_, Fe_3_Mn_3_O_8_, and MnCr_2_O_4_ when combined with XRD data shown in Figure 6. Mn- and Cr-depleted areas formed in the scale–substrate interface area, which are now filled with a high concentration of Ni, Co, and Fe. Kim et al. [13] reported that a decrease in elements in the interface region may reduce the sluggish diffusion effect, which further influences high-temperature oxidation resistance. From EDS analysis, it can be confirmed that the oxide scales are divided into two layers, including the manganese-rich outer layer and the chromium-rich inner layer. The distribution of elements is correlated with the oxygen affinities of Mn and Cr as well as the outward diffusion rates of cations in the oxide scale [20].

The oxidation behaviour of the CrMnFeCoNi HEA provides valuable insights into its suitability for high-temperature applications. This oxidation behaviour of CrMnFeCoNi HEA plays a critical role in determining its mechanical, thermal, and corrosion resistance properties in elevated temperature environments. The observed non-protective oxide formation, particularly the depletion of Mn in the bulk alloy, can have significant implications for this particular alloy’s mechanical properties. Mn is known to contribute to solid solution strengthening and can enhance the mechanical strength and ductility of alloys. Therefore, the formation of non-protective oxides and the depletion of Mn may negatively affect the mechanical performance of the CrMnFeCoNi HEA at high temperatures.

### 4.4. Oxidation Thermodynamics

The computed oxide phases, as a function of oxygen partial pressure for CrMnFeCoNi alloy at starting equilibrium conditions of 1 atm (1.013 × 10^5^ Pa) and 1000 °C, are presented in Figure 10a. It can be seen that the total oxides that formed at the beginning are spinel. With the decrease in oxygen partial pressure, the fraction of the spinel starts to decrease from unity at 3.42 × 10^−2^ atm. The volume fraction of the spinel oxides decreases linearly until it reaches 0.5080 volume fraction at 1.47 × 10^−10^ atm. At the beginning of the oxidation process at atmospheric pressure, no other form of oxides formed, but as the oxygen partial pressure drops below 0.0342 atm, the halite type of oxide compensates for the decrease in spinel oxide. At the same oxygen partial pressure of 1.47 × 10^−10^ atm, the halite form of oxides reaches its maximum value of 0.492. As can be seen from Figure 10a, the value of partial pressure 1.47 × 10^−10^ atm is the critical value for both oxide types, at this point chemically different oxide of the same phase type forms. The new spinel oxide starts to decrease, and halite increases gradually until oxygen partial pressure drops to 1 × 10^−14^ atm.

Figure 10b shows the elemental composition of spinel oxides as a function of oxygen partial pressure. Ni does not form spinel oxide and remains at a lower level at all pressure values, which agrees with the experimental results where no Ni-containing oxides were observed. ThermoCalc predicts Cr and Co to be major elements in the formed spinel oxide type at atmospheric pressure, but with decreasing oxygen partial pressure, the Cr amount in the spinel oxide increases while Co decreases. The Mn amount remains almost the same (i.e., 1.12 × 10^−1^ and 1.213 × 10^−1^ atm) in spinel throughout with pressure drop. Similarly, Figure 10c shows the elemental analysis of halite form of oxides. As halite started to form at 3.42 × 10^−2^ atm at a lower oxygen partial pressure, so the elements amount is zero at the beginning. ThermoCalc predicts that in a vacuum at comparatively high pressure, Ni will preferentially form a higher amount of halite followed by Co, and Cr will not participate in halite formation. At a lower pressure (1.0 × 10^−14^ atm), Mn makes up a large proportion of halite oxide.

Figure 10a shows the ThermoCalc calculations of oxide scale formations on a CrMnFeCoNi HEA and could be used to explain the mechanism for the formation of multi-layered oxides on as-prepared CrMnFeCoNi HEA after oxidation, as shown earlier in Figure 5c and Figure 7c. HTXRD of CrMnFeCoNi HEA investigated the oxidation behaviour in two different environments: vacuum (4.2 × 10^−6^ atm) and air (1.0 atm). ThermoCalc predicts that halite begins to form at pressure *p*(O_2_) of 3.42 × 10^−2^ atm with the majority of NiO type which co-exists with other spinel oxide types. However, halite oxide in the form of MnO was the only oxide type that formed during HTXRD in a vacuum (4.2 × 10^−6^ atm) at 700 °C. ThermoCalc predicts NiO as the major halite oxide type even at the same equilibrium temperature (700 °C) at which the initial oxide MnO formed during an experiment in a vacuum. In a review of free energies of formation of possible oxides at 950 °C [20] (ΔGf: in kJ/mol. O_2_), the ΔG_f_ of MnO (−590.5) is much more negative than that of NiO (−260.4). Based on the free energies of oxide formation, the found oxide types agree well with the oxide type that could form regarding free energies. Also, ThermoCalc predicts other halite oxide types of Co which were not observed during the experiment. Such a discrepancy in results between the ThermoCalc-predicted oxide types (i.e., NiO, CoO) and the experimentally found one (MnO only) could be the time scale, which means a sufficiently long time for the system to attain its most stable equilibrium state.

Similarly, ThermoCalc predicts spinel oxide formation at normal atmospheric pressure and 1100 °C temperature. Meanwhile, during HTXRD in air, corundum oxides formed initially at 550 °C, which is different from what ThermoCalc predicted. The formation of corundom (Mn_2_O_3_) during the experiment can be explained with the increase in temperature, and the initially formed corundom oxide changed to spinel (Mn_3_O_4_) at a higher temperature (950 °C). If considering oxides that form at a higher temperature such as 950, 1000, and 1050 °C, the ThermoCalc-predicted spinel oxide type is the same as the experimentally found spinel oxide. However, ThermoCalc predicts Co as a second major element in the spinel oxide that can form, but during the experiment, spinel oxides contain only Mn, Cr, and Fe (i.e., Mn_3_O_4_, MnCr_2_O_4_, and Fe_3_Mn_3_O_8_). ThermoCalc’s predicted oxide structure and chemical composition at a high temperature and in ambient air are closer to the experimentally found results than those in the vacuum. In order to further advance our understanding of oxidation kinetics and the evolution of the oxide layer, future research should consider integrating detailed characterisation techniques such as TGA and DTA. Additionally, incorporating in situ microscopy or spectroscopy methods would offer real-time information, allowing for a more comprehensive analysis of the oxidation kinetics and the dynamic changes in the oxide layer. These additional techniques would significantly contribute to unravelling the intricate processes involved in alloy oxidation.

## 5. Conclusions

In this work, an equiatomic CrMnFeCoNi HEA was successfully fabricated through the arc-melting process, with an emphasis on phase, microstructure, and real-time oxidation behaviour in two different non-ambient (high-temperature and low-oxygen partial pressure) conditions. The main findings of this research work can be summarised as follows.

In a vacuum (4.2 × 10^−6^ atm), only MnO, a halite-type oxide, was formed during in situ oxidation, contrary to ThermoCalc’s prediction of both halite and spinel oxides, and Ni/Co forming halite.In air (1.0 atm), vigorous oxidation was observed, with most of the oxides formed being spinel. Mn remained important in the formation of spinel oxides, and Co was predicted to form spinel oxides by ThermoCalc but was not found in oxide-forming elements. Oxides like Cr_2_O_3_, Fe_2_O_3_, and Fe_3_O_4_ were not detected during in situ HTXRD in either vacuum or air, which contrasts with previous studies on the isothermal oxidation of CrMnFeCoNi HEA.Mn was found to be a higher oxide-forming element, producing non-protective oxides, and leading to the depletion of Mn in the bulk alloy beneath the oxide layer. Mn depletion creates doubt about the application of HEA for high-temperature applications and due to the higher diffusion rate may affect the sluggish kinetic diffusion effect, one of the four characteristic features of HEAs.

## Figures and Tables

**Figure 1 materials-16-05042-f001:**
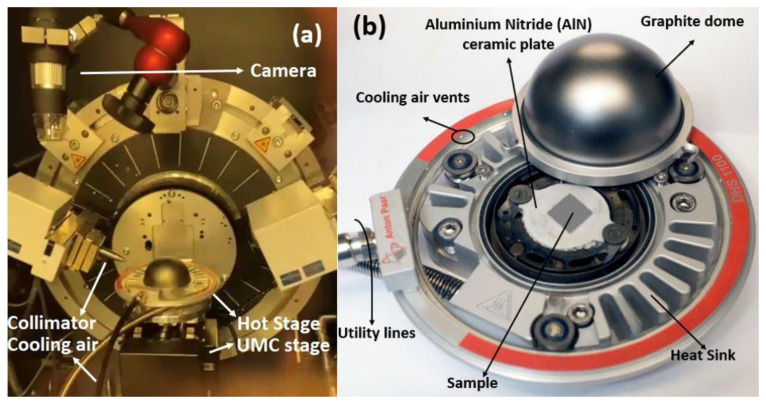
(**a**) HTXRD experiment set-up in Bruker D8 Advance (Karlsruhe, Germany); (**b**) Anton Paar Hot Stage for non-ambient XRD.

**Figure 2 materials-16-05042-f002:**
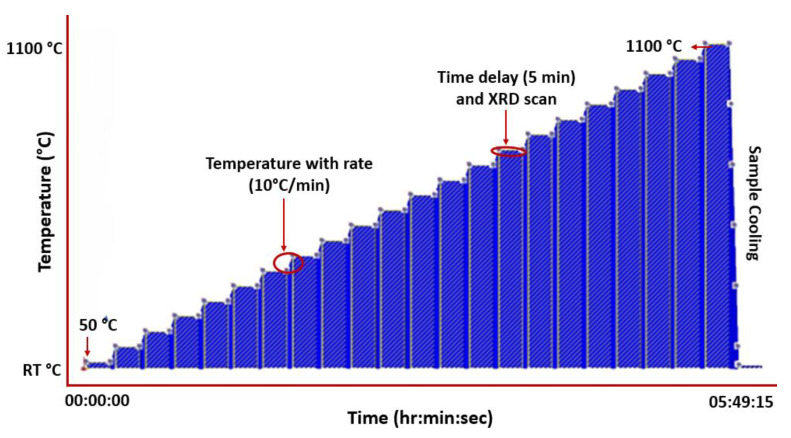
Temperature profile set-up for HTXRD of CrMnFeCoNi HEA in air and vacuum.

**Figure 3 materials-16-05042-f003:**
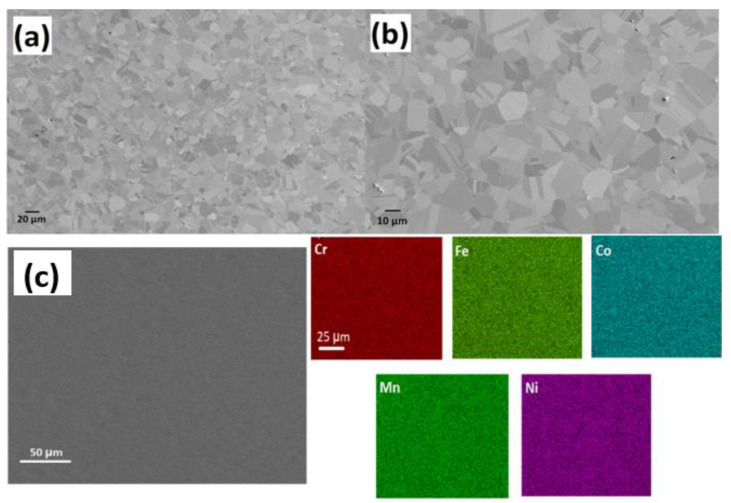
SEM images (**a**,**b**) shows the microstructure of CrMnFeCoNi HEA at different magnifications; (**c**) EDS mapping of elements shows sample elements distribution as prepared HEA.

**Figure 4 materials-16-05042-f004:**
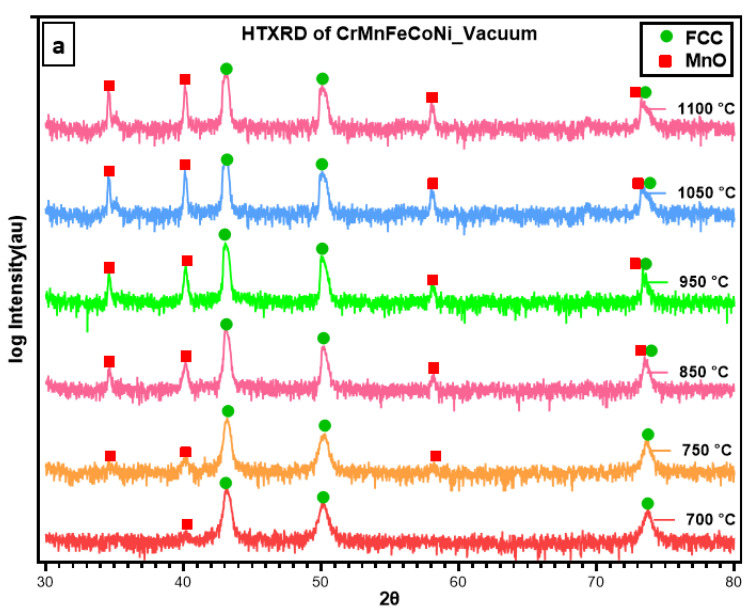
(**a**) HTXRD scans of CrMnFeCoNi HEA in a vacuum showing phase formation as alloy starts to oxidize; (**b**) a densitometric view of HTXRD recorded during heating from room temperature (RT) to 1100 °C.

**Figure 5 materials-16-05042-f005:**
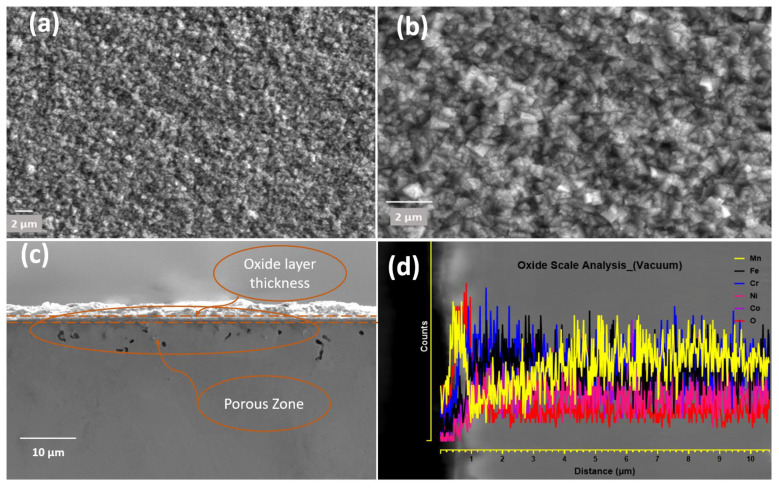
CrMnFeCoNi HEA oxidized during HTXRD in a vacuum: (**a**) oxide surface; (**b**) oxide surface at higher magnification; (**c**) cross-section of oxide layer showing oxide scale with porosity beneath oxide surface; (**d**) EDS line-scale analysis of oxide thickness.

**Figure 6 materials-16-05042-f006:**
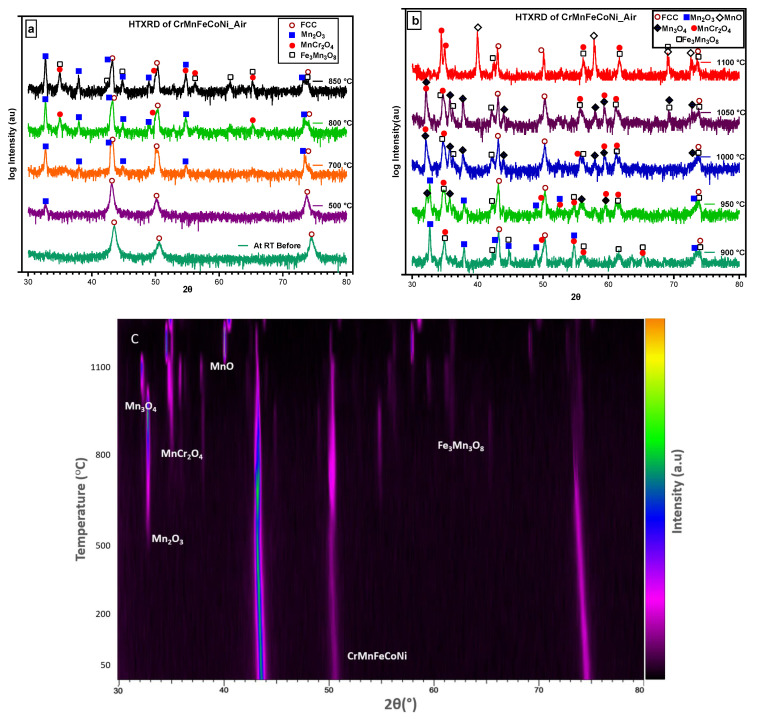
Selected diffractograms of HTXRD of CrMnFeCoNi HEA: (**a**) the initiation and growth of the first oxide formed; (**b**) different oxides formed with increase in temperature; (**c**) oxides formed and the initiation of new oxides and consumption of earlier formed oxides.

**Figure 7 materials-16-05042-f007:**
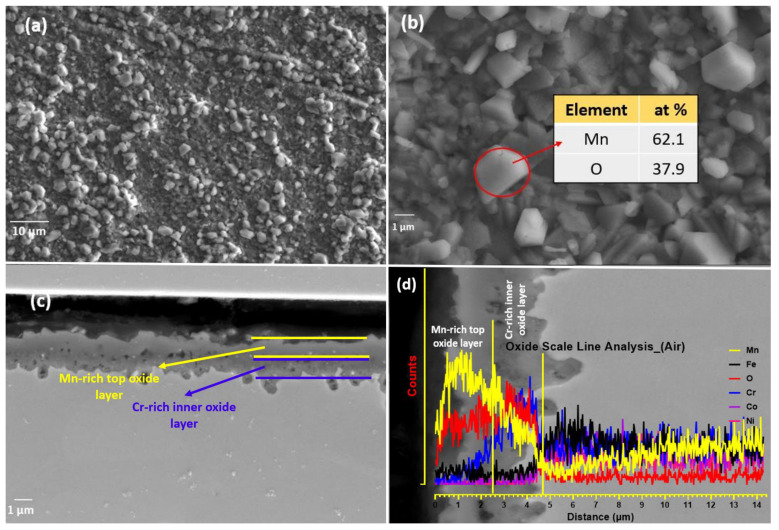
(**a**) Oxidized surface of CrMnFeCoNi HEA during HTXRD in air; (**b**) granules of manganese oxide on surface; (**c**) cross-section of the oxide layer developed in air; (**d**) EDS line-scale of oxide layer with porous structure.

**Figure 8 materials-16-05042-f008:**
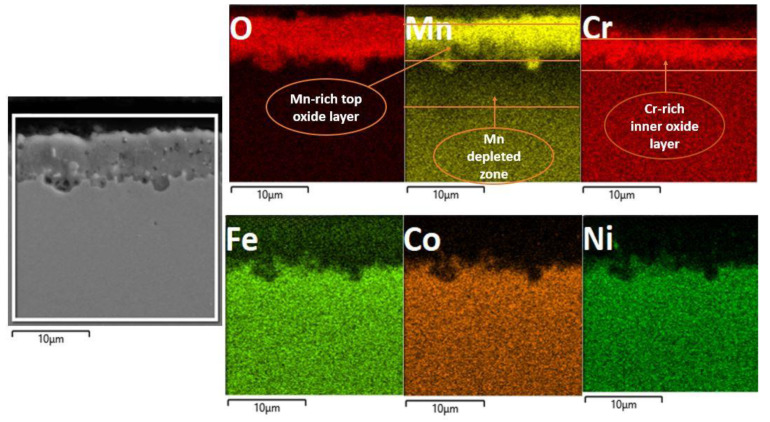
EDS map analysis of the oxide layer shows the multi-layered oxide structure, with a top Mn-rich layer an inner Cr-rich oxide layer and followed by Mn depleted zone.

**Figure 9 materials-16-05042-f009:**
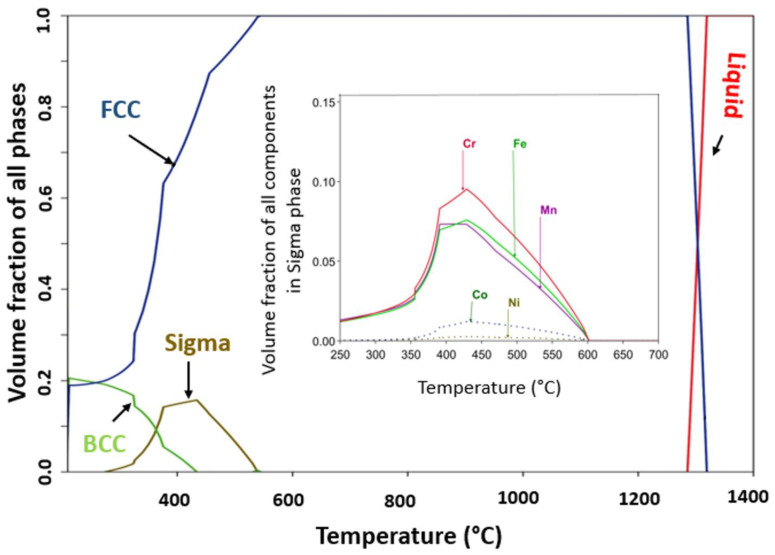
CALPHAD-predicated temperature vs. phase fraction phase diagram for CrMnFeCoNi HEA. Inset is the CALPHAD predicted sigma phase formed at a lower temperature.

**Figure 10 materials-16-05042-f010:**
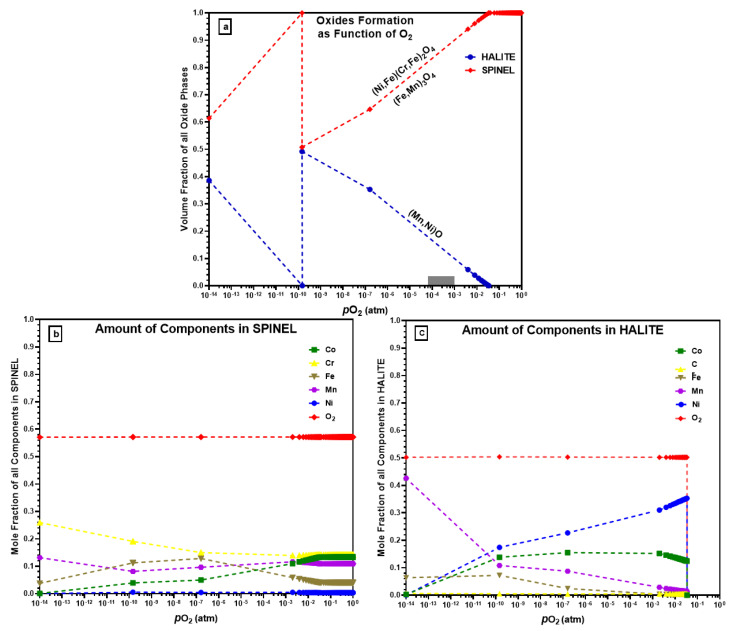
(**a**) Types of oxides that form using ThermoCalc; (**b**) elements in the formation of spinel oxides; (**c**) elements in the formation of halite oxides.

**Table 1 materials-16-05042-t001:** Nominal and EDS chemical composition of CrCoFeMnNi, the HEA used in this study.

Alloy Composition (at.%)	Co	Cr	Mn	Fe	Ni
Nominal	20	20	20	20	20
Experimental (EDS)	21.04	18.22	20.34	20.37	20.38

**Table 2 materials-16-05042-t002:** Initialisation parameters for ThermoCalc calculations.

Temperature (°C)	1100	
Pressure (Pa)	101,325	
System Size (mol)	1.0	
Composition	Cr	15
	Mn	15
	Fe	15
	Co	15
	Ni	35
	O	05
Activity of O_2_	Min	1.0 × 10^−14^
	Max	1.0
	No of Steps	100

**Table 3 materials-16-05042-t003:** The experimental data and CALPHAD predictions for long-term annealed CrMnFeCoNi HEA.

	Composition	Homogenisation Conditions	Ageing Conditions	Observed Phases	CALPHAD Phases	Solvus	Equili Solidus
Otto et al. [38]	CrCoFeMnNi	1200 °C 48 h	500 °C 12,000 h	L1_0_, BCC, B2	FCC, Sigma	798	1286
	CrCoFeMnNi	1200 °C 48 h	700 °C 12,000 h	FCC, Sigma	FCC, Sigma	798	1286
	CrCoFeMnNi	1200 °C 48 h	900 °C 12,000 h	FCC	FCC	798	1286
Pickring et al. [36]	CrCoFeMnNi	1240 °C 1000 h	700 °C 1000 h	FCC, Sigma, M_23_C_6_	FCC, Sigma	798	1286

## Data Availability

This is a research article, and results after data analysis are presented within the article. The data presented in the figures are available on request from the corresponding author.

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
