# Peer review of "Thermodynamic Insights into the Oxidation Mechanisms of CrMnFeCoNi High-Entropy Alloy Using In Situ X-ray Diffraction"

_materials, 2023, doi:10.3390/ma16145042_

Round 1
Reviewer 1 Report
The manuscript discusses the effect of temperature on oxidation of a CrMnFeCoNi High Entropy Alloy. The study is in detail and helpful to understand the phase transition of such alloys at different temperatures and key element responsible for the oxidation. I have some minor comments as below:- Figure 4(a) shows the HTXRD where, we see the emergence of the MnO peak at 40 deg 2 theta angle which is not marked at T = 700 C. Ideally, the oxidation starts at 700 C and should be indicated there.
- Since, HTXRD is the key characterization for this study, the authors should use log scale for all the XRD patterns shown in the paper. This will help to visualize that no additional oxide phases are present other than the ones detected by the authors.
Reviewer 2 Report
Overall, the paper presents a well-executed study with detailed experimental observations and analysis, contributing to the understanding of oxidation processes in CrMnFeCoNi. The findings raise important questions regarding the formation of protective oxides and suggest future research.
-The introduction should provide more context and background information on CrMnFeCoNi High Entropy Alloy (HEA) (magnetic properties for example) and its high-temperature oxidation behavior. Discuss the significance of studying CrMnFeCoNi, their potential applications, and more existing literature on their oxidation behavior.
-The comparison of experimental results with thermodynamic calculations using ThermoCalc is an interesting approach. However, it is necessary to provide more information about the specific parameters, assumptions, and limitations of the thermodynamic calculations. Additionally, discuss more discrepancies between the experimental observations and the thermodynamic predictions, and the potential factors that may contribute to these differences.
-While the study mentions the significance of the findings in terms of the development of high-performance alloys, it would be helpful to discuss the potential implications of the results in specific applications. Address with more details the impact of the observed oxidation behavior on the mechanical, thermal, magnetic or corrosion properties of the CrMnFeCoNi HEA and its suitability for high-temperature applications.
-The findings related to the major oxide-forming element, Mn, in both vacuum and air environments, and the absence of oxides like Cr2O3, Fe2O3, and Fe3O4 are significant. However, more details about the specific experimental observations, such as the oxidation kinetics and the evolution of the oxide layer, would provide a more comprehensive understanding of the results.
-Authors said:” The reason of forming ternary oxides instead of binary may be that stable binary oxides of Cr and Fe would form after long exposure time at the oxidation temperature.” Authors should better explain this statement.
-The study provides valuable insights into the oxidation mechanisms of CrMnFeCoNi HEA using in-situ XRD, demonstrating the power of this technique in understanding phase transformations and oxidation behaviors in complex alloys.
